# Does the Chimerization Process Affect the Immunochemical Properties of WNV-Neutralizing Antibody 900?

**DOI:** 10.3390/ijms262412181

**Published:** 2025-12-18

**Authors:** Anastasiya A. Isaeva, Valentina S. Nesmeyanova, Daniil V. Shanshin, Nikita D. Ushkalenko, Ekaterina A. Volosnikova, Tatiana I. Esina, Elena V. Protopopova, Victor A. Svyatchenko, Valery B. Loktev, Sergey E. Olkin, Elena D. Danilenko, Elena I. Kazachinskaia, Dmitriy N. Shcherbakov

**Affiliations:** State Scientific Center of Virology and Biotechnology “Vector”, Rospotrebnadzor, 630559 Koltsovo, Russia; nesmeyanova_vs@vector.nsc.ru (V.S.N.); shanshin_dv@vector.nsc.ru (D.V.S.); ushkalenko_nd@vector.nsc.ru (N.D.U.); volosnikova_ea@vector.nsc.ru (E.A.V.); esina_ti@vector.nsc.ru (T.I.E.); protopopova_ev@vector.nsc.ru (E.V.P.); svyat@vector.nsc.ru (V.A.S.); loktev@vector.nsc.ru (V.B.L.); olkin@vector.nsc.ru (S.E.O.); danilenko_ed@vector.nsc.ru (E.D.D.); lena.kazachinkaia@mail.ru (E.I.K.); scherbakov_dn@vector.nsc.ru (D.N.S.)

**Keywords:** monoclonal antibodies (mAbs), West Nile virus (WNV), chimerization, Fc engineering, antigen–antibody complex

## Abstract

West Nile fever is an infectious disease caused by the West Nile virus (WNV), which is transmitted by mosquitoes. Epidemiological surveillance confirms the potential risk of WNV infection in human populations. The lack of specific antiviral therapeutics and vaccines against WNV underscores the urgent need to develop effective therapeutic approaches. In this study, a recombinant chimeric monoclonal antibody (mAb) 900 was generated based on the broadly neutralizing and protective murine mAb 9E2. The antigen-binding regions of the murine mAb were fused with the constant domains (CH2–CH3) of human IgG1. Two key amino acid clusters, M252/S254/T256 and H433/N434, were introduced into the CH2–CH3 domains to enhance the affinity of mAb 900 for the neonatal Fc receptor (FcRn). The engineered mAb 900 was produced in CHO cells and purified to high homogeneity. Biophysical characterization confirmed its stability and correct dimeric assembly. Comparative analysis demonstrated that mAb 900 retained the high antigen-binding affinity and potent virus-neutralizing activity of its murine predecessor. Most importantly, mAb 900 demonstrated significant protective efficacy in a lethal mouse model of WNV infection. These results establish the proof of concept for mAb 900 as a promising candidate for further preclinical development against WNV infection.

## 1. Introduction

The West Nile virus (WNV) is an orthoflavivirus belonging to the Japanese encephalitis antigenic complex and transmitted by mosquitoes [1,2,3]. According to the World Health Organization, WNV is currently endemic in Africa, the Middle East, the United States, Australia, Europe, and Asia, reflecting its capacity for global dissemination [4,5,6]. In 2024, the United States reported the highest number of laboratory-confirmed West Nile fever (WNF) cases, totaling 1466 infections, which represents a 1.8-fold decrease compared to the number of cases in 2023 [6]. Nevertheless, the long-term average incidence in the United States remains high, with an annual mean of 2234 cases recorded between 1999 and 2023. In Europe, 1437 cases were reported, primarily in Central and Mediterranean regions, with Italy accounting for the largest number (455 cases) [6]. In Asia and Australia, WNF cases are sporadic, and most Asian countries lack mandatory epidemiological surveillance for this infection [6,7]. A consistent expansion of WNV-endemic areas has been observed. Between 1999 and 2020, WNV infections were documented in 51 regions of the Russian Federation. Epidemiological analysis for 2024 revealed a 2.1-fold increase in cases compared with 2023 (440 vs. 210 cases) [6]. The rising frequency of outbreaks and the emergence of infections in temperate regions indicate the evolving epidemiology of orthoflaviviruses and their capacity to adapt to new environmental conditions [4].

The absence of specific therapeutic agents and vaccines against WNV underscores the urgent need to identify effective countermeasures [1,8,9]. Among the most promising tools in combating viral infections are monoclonal antibodies (mAbs) [10,11,12]. For instance, the mAb Palivizumab, developed for the prevention of severe respiratory syncytial virus infection in children, has demonstrated both efficacy and safety and is currently approved for use in more than 70 countries worldwide [13,14]. Neutralizing antibodies targeting the WNV envelope (E) protein have been shown to inhibit WNV infection both in cell culture and in animal models [8,15]. Therefore, ongoing studies are focused on developing mAbs with enhanced efficacy against WNV.

Despite their high therapeutic potential, the clinical application of mAbs is associated with several challenges. One of the major issues is the adaptation of heterologous antibodies that exhibit promising neutralizing activity against viral pathogens. The high immunogenicity of heterologous antibodies can trigger undesirable immune responses, such as the production of antibodies against the drug [16]. These responses may compromise therapeutic efficacy and increase the risk of adverse effects, including allergic reactions. To minimize these risks, antibody humanization strategies have been developed [17]. The humanization process involves substituting amino acid residues in heterologous antibodies with their human counterparts to improve compatibility with the patient’s immune system and reduce immunogenicity [18]. However, this process carries a risk of diminished functional activity, as such substitutions may alter the three-dimensional structure of the complementarity-determining regions and weaken antigen binding affinity. Studies have demonstrated that even minor modifications within these critical regions can significantly reduce antigen binding and, consequently, therapeutic efficacy [17,19,20]. A well-established alternative approach is the development of chimeric antibodies, which are constructed by fusing the variable regions of the parental animal antibody with the constant (Fc) regions of a human antibody. This strategy preserve the neutralizing activity of the parental antibody while reducing immunogenicity [20]. The chimerization of antibodies, which involves the replacement of constant domains, is generally regarded as more predictable than humanization, a process that requires modifications within the variable regions. Nevertheless, even chimerization is not fully predictable. The substitution of constant domains can disrupt interdomain interactions and alter the overall antibody conformation, potentially compromising its stability and antigen-binding affinity. Consequently, a thorough characterization of any chimeric antibody must include a comparative assessment of its immunochemical properties against those of the parental murine antibody.

The objective of this study was to engineer a chimeric, neonatal Fc receptor (FcRn) -enhanced mAb 900 and to evaluate its potential against WNV through an analysis encompassing *in vitro* immunochemical characterization and *in vivo* protective efficacy.

## 2. Results

### 2.1. Fc Fragment Engineering of Monoclonal Antibody (mAb) 900

In this study, we generated a recombinant antibody in a single-chain variable fragment (scFv) fused to the Fc region of a human immunoglobulin (scFv-Fc format). To enhance its serum half-life, the Fc domain was engineered with specific amino acid substitutions known to improve binding affinity to the FcRn. We designed and constructed four variants (V1–V4) of the anti-WNV antibody 900, each with a different Fc composition (Figure 1a,b). The variable regions were derived from the murine mAb 9E2 (Figure 1c), which was previously generated at the State Research Center of Virology and Biotechnology “Vector” using hybridoma technology. This broadly reactive mAb 9E2 recognizes the WNV E protein in immunoblot analysis and specifically binds WNV in an enzyme-linked immunosorbent assay (ELISA) with a titer of at least 1:2,187,000. Moreover, mAb 9E2 is capable of neutralizing multiple WNV strains and protecting animals from lethal infection [21,22,23].

The constructed expression vectors were used to establish stable producer cell lines derived from CHO-K1 cells. As a result, four stable CHO-K1-900 producer cell lines were obtained, each secreting one of the four chimeric mAb 900 variants (V1–V4, Figure 1b). Each antibody variant was produced at a yield of approximately 10 mg. Purification was carried out by affinity chromatography using Protein A columns.

Recombinant FcRn produced in mammalian cells was employed to evaluate the binding affinity of the mAb 900 variants (see Section 4.2, Appendix A). Interactions between the purified antibody variants and recombinant FcRn were analyzed using biolayer interferometry (BLI), a technique that enables direct real-time measurement of protein–protein binding kinetics and affinity without the need for labeling [24,25].

Analysis of the equilibrium dissociation constants (KD) revealed distinct differences in FcRn binding affinity among the four mAb 900 variants (Table 1).

Variant V4, incorporating both mutation clusters (M252/S254/T256 and H433/N434), exhibited the lowest KD, indicating the highest binding affinity to FcRn (mean KD = 1.41 × 10^−9^ M). In contrast, variant V1, which lacked these mutations, displayed the highest KD, consistent with weak FcRn binding.

These findings demonstrate that the strategic introduction of mutations into both critical Fc fragment clusters of mAb 900 substantially enhanced its affinity for FcRn.

### 2.2. Cultivation of the CHO-K1-900 Cell Line and Purification of Chimeric mAb 900

Based on the FcRn binding affinity assessment, variant V4 of mAb 900, incorporating both mutation clusters (M252/S254/T256 and H433/N434) in the Fc fragment, was selected for subsequent experiments.

The selected mAb 900 variant was produced by cultivating the CHO-K1-900 cell line. The initial seeding density was 1 × 10^6^ cells/mL, and cells were maintained at 37 °C. Cultivation continued until the cell density reached 6–8 × 10^6^ cells/mL, after which the culture was shifted to hypothermic conditions (31 °C) for further growth. Cultivation was carried out over 14 days, with continuous monitoring of cell viability and maintenance of glucose concentration at 40 mM. Under these conditions, the strain achieved a productivity of 200 mg/L of culture medium.

mAb 900 was purified using a multistep procedure combining affinity and ion-exchange chromatography (see Section 4.5). This purification strategy yielded the target protein with >95% purity, as confirmed by SDS-PAGE, Western blot, reversed-phase high-performance liquid chromatography (RP-HPLC) (Figure 2) and size-exclusion chromatography (SEC) (Appendix A).

Western blot analysis under reducing conditions using an anti-human Fc-specific antibody confirmed the identity of mAb 900, revealing a predominant band at approximately 50 kDa. This corresponds to the expected molecular weight of the denatured, monomeric polypeptide chain (calculated: 50.8 kDa) and validates successful expression. An additional high-molecular-weight band (>200 kDa), also detected by both reducing SDS-PAGE and RP-HPLC (Peak 3), indicated the presence of a minor fraction of antibody aggregates.

The oligomeric state and purity of the native protein were assessed by SEC. The SEC profile showed a single major peak, corresponding to a molecular mass of 100.0 ± 0.5 kDa, which is consistent with the dimeric scFv-Fc format of mAb 900 (Appendix A, Appendix A). This confirmed that the purified preparation was predominantly in the correct, monodisperse form.

Finally, the conformational stability of mAb 900 was evaluated by monitoring its thermal unfolding via circular dichroism (CD) spectroscopy. The CD thermal denaturation profile exhibited two distinct transitions at 59 °C and 77 °C, with the higher transition (77 °C) representing the complete unfolding of the protein’s native structure, indicating a stable fold (Appendix A).

### 2.3. Functional Activity of mAb 900 in Enzyme-Linked Immunosorbent Assay (ELISA)

The functional activity of mAb 900 was evaluated using ELISA to determine its binding capability to antigens of inactivated WNV (strain LEIV-Vlg99-27889-human) and the recombinant D III domain of the WNV structural E glycoprotein.

Comparison of the chimeric mAb 900 with the original murine mAb 9E2 in assays using inactivated virus revealed differences in their titers (Figure 3a). To ensure an accurate comparison, titers were normalized based on protein concentration per milliliter in each sample (Table 2).

The murine mAb 9E2, used as a reference control, has been previously characterized to bind the WNV E protein, neutralize a broad panel of WNV strains (LEIV-Vlg00-27924-human, LEIV-Vlg99-27889-human, LEIV-Az67-1640-nuthatch, LEIV-Az70-72-ticks, LEIV-Tur73-2914-ticks, Ast63-94-ticks, NY99, and Egypt 101), and protect animals from lethal challenge [21,22,23].

The normalized titer of mAb 900 in the culture supernatant was 4.20 lg, whereas the titer of the purified mAb 900 was 4.61 lg. The increase in titer may be attributed to the removal of impurities present in the culture supernatant, which could have interfered with the binding of mAb 900 to the antigen, thereby affecting the final antibody titer.

The murine mAb 9E2 displayed a higher titer (4.64 lg). The slightly reduced functional activity (not statistically significant) of the chimeric antibody in ELISA compared with its murine counterpart may result from structural modifications, such as alterations in the Fc fragment, which can affect antigen binding.

ELISA using the recombinant D III domain of the WNV E glycoprotein demonstrated endpoint titers 5.8 lg, for mAb 9E2, 4.9 lg for mAb 900. These findings are consistent with the results obtained using inactivated WNV, which similarly showed a higher titer for mAb 9E2 compared with the chimeric mAb 900 (Figure 3b).

### 2.4. Virus-Neutralizing Activity of mAb 900

The results of the neutralization assay of mAb 900 against WNV (LEIV-Vlg99-27889-human) are summarized in Table 3.

Murine mAb 9E2, previously shown to possess neutralizing activity [21,22,23], was used as a control. The IC_50_ of purified mAb 900 was 74 ng/mL, 2.5-fold lower than that of unpurified mAb 900 (185 ng/mL), indicating a substantial enhancement of the antibody’s specific functional activity following removal of impurities. This improvement likely reflects the elimination of components that hinder specific antigen binding.

Comparison of the IC_50_ values of the chimeric mAb 900 (74 ng/mL) and the murine mAb 9E2 (68.6 ng/mL) indicates that the chimerization had minimal impact (not statistically significant) on the antibody’s neutralizing activity. This finding demonstrates that the broadly reactive murine mAb 9E2’s neutralizing properties were largely preserved in mAb 900.

### 2.5. Equilibrium Dissociation Constants (KD) Estimate for mAb 900

Biolayer interferometry was used to assess the antibody’s affinity to the antigen. MAb 9E2 interacts with recombinant DIII protein, of the WNV surface protein, with an affinity of ~1.51 × 10^−9^ M (Figure 4a). Similarly, recombinant mAb 900 strongly binds DIII with a KD of ~2.30 × 10^−9^ M (Figure 4b).

From comparing the obtained values using the Dunnett test, the difference between the values was statistically not significant (*p*-value > 0.5). This means that the affinities of mAb 9E2 and 900 for antigen are essentially equivalent.

### 2.6. Glycosylation Profile of mAb 900

The results of the N-glycan profile analysis for mAb 900 samples, determined by hydrophilic interaction liquid chromatography (HILIC-HPLC), are provided in Appendix A, Appendix A and Appendix A. The analysis was performed on two mAb 900 samples (sample 1 and sample 2) produced under identical conditions with a two-month interval.

The following types of neutral N-linked glycans were identified: G0, G0F, G0-N, G0F-N, G1, G1F, G2F, and Man5. The content of afucosylated glycans (AF, %) was approximately 7%. The content of galactosylated glycans (G, %) varied between the samples: 37.6% for sample 1 and 47.3% for sample 2. This variation is attributed to a redistribution in the relative abundance of the major glycoforms G0F and G1F. The degree of mannosylation was 0.9% and 0.5% for sample 1 and sample 2, respectively.

### 2.7. Assessment of Acute Toxicity in Mice

An acute toxicity study was performed by administering a single high dose (200 µg per mouse) of mAb 900 to healthy ICR mice (n = 6) via intraperitoneal injection. All animals survived the 14-day observation period with no signs of adverse effects, changes in body weight, or altered behavior compared to the control group. These results indicate the absence of acute toxicity for mAb 900 at the tested dose.

### 2.8. Protective Efficacy of mAb 900 In Vivo

The results assessing the protective activity of the recombinant chimeric mAb 900 against infection with the WNV strain LEIV-Vlg99-27889-human (0.3 × 10^2^ LD_50_ per mouse) are presented in Table 4 and Figure 5. The experiment was conducted using a BALB/c weanling mouse model, with mAb 900 administered 6 h post-infection.

A comparison of the therapeutic efficacy of different mAb 900 doses revealed that the 100 µg per mouse dose provided the highest level of protection (83% survival) against the challenge dose of 30 LD_50_ WNV. For the group receiving the 100 µg dose, the survival curve was statistically significantly different from the virus control group (χ^2^ = 6.721, df = 1, *p* = 0.0095), indicating a pronounced protective effect of the antibody at this dose. Doses of 50 µg and 200 µg per mouse protected 33% of the animals. No statistically significant difference in survival compared to the virus control was observed for the groups receiving 50 µg (χ^2^ = 0.2198, df = 1, *p* = 0.6392) or 200 µg (*p* > 0.05). Thus, a single dose of 100 µg of mAb 900 per animal is capable of exerting protective effects against WNV *in vivo*.

## 3. Discussion

Antibodies are central to adaptive immunity, preventing pathogen dissemination and providing long-term protection, which is critical in combating viral infections [27]. Their ability to directly suppress viral replication makes them a compelling modality for both therapeutic and prophylactic applications. Broadly neutralizing antibodies, which exhibit high potency against diverse viral strains, are of particular interest. Such antibodies are often first identified in animal models; however, their direct therapeutic use in humans is limited by immunogenicity [18,20]. Antibody humanization strategies, including chimerization, have been developed to overcome this hurdle. A paramount concern in this process is that replacing heterologous constant regions can unpredictably alter the structural integrity and affinity of the antigen-binding site [28]. Therefore, the primary objective of this study was to generate a humanized, Fc-engineered variant of the broadly neutralizing murine anti-WNV antibody 9E2 [22,23] and to perform a rigorous, head-to-head comparison of its key functional and biophysical properties.

To enhance the therapeutic profile of the candidate, we employed rational Fc engineering. We grafted the variable regions of mAb 9E2 onto a human IgG1 scaffold incorporating the M252Y/S254T/T256E and H433K/N434F mutations. These mutations are well-documented to increase affinity for the neonatal FcRn receptor, a key mechanism for extending serum half-life and improving drug exposure [29,30]. Our BLI analysis confirmed the success of this design, demonstrating high-affinity binding of the engineered Fc (variant V4) to human FcRn (KD = 1.41 × 10^−9^ M). This strategic modification positions mAb 900 for improved pharmacokinetics, aligning with clinically validated approaches for other antiviral antibodies [29].

The most critical finding of this work is that the extensive molecular engineering—encompassing chimerization and targeted Fc mutagenesis—did not compromise the core antiviral function inherited from the parental antibody. Our comparative functional analysis revealed near-identical profiles between mAb 900 and murine 9E2. The chimeric antibody retained high-affinity binding to the WNV E protein DIII domain (KD = 2.30 × 10^−9^ M), comparable to the progenitor (KD = 1.51 × 10^−9^ M). Most importantly, its potent virus-neutralizing activity *in vitro* remained fully intact, with virtually identical IC_50_ values (74 ng/mL vs. 68.6 ng/mL). These values align with the high-affinity range (10^−9^–10^−11^ M) reported for other therapeutic antibodies against flaviviruses [31,32,33,34] and confirm the structural preservation of the antigen-binding site. This successful retention of function post-humanization places mAb 900 among other successful cases where chimerization preserved key properties [18,20,28].

Beyond function, a comprehensive developability assessment underscores the robustness of mAb 900. The antibody exhibits high conformational stability, with a thermal denaturation profile showing a major unfolding transition at 77 °C, consistent with stable, well-folded human IgG1 therapeutics. SEC confirmed the preparation was predominantly monomeric (>92%), and a detailed glycosylation analysis revealed a typical profile for CHO-derived antibodies. The low levels of afucosylation (~7%) and high-mannose glycans (<1%) [35,36,37] are characteristic of non-engineered production systems and predict predictable effector function and pharmacokinetic behavior. Prior to *in vivo* efficacy testing, an acute toxicity study in ICR mice confirmed the absence of adverse effects, establishing an initial safety profile for the formulated product.

The most significant advance presented here is the demonstration of *in vivo* protective efficacy. In a lethal challenge model using a neuroinvasive WNV strain, a single, post-exposure dose of mAb 900 (100 µg) conferred significant protection, with 83.3% survival. This result provides proof-of-concept for its potential as promising preclinical candidate. The dose–response relationship observed—with lower efficacy at both higher (200 µg) and lower (10, 50 µg) doses—warrants further investigation. The complete lethality at the 10 µg dose, occurring earlier than in the virus control group, raises the possibility of ADE, a phenomenon that must be carefully ruled out for flavivirus therapeutics during future development. This complex efficacy profile highlights the need for detailed future studies to optimize the dosing regimen.

## 4. Materials and Methods

### 4.1. Construction of the Integration Vector pVeal2_9E2ch

The nucleotide sequences of the 9E2 heavy (GenBank ABU63295.1) and light (GenBank ABU63296.1) chain genes were retrieved from the database and codon-optimized for CHO cells using the Gene Optimizer tool (https://www.thermofisher.com/ru/en/home/life-science/cloning/gene-synthesis/geneart-gene-synthesis/geneoptimizer.html (accessed on 2 February 2020)). The final 9E2ch-scFv nucleotide sequence was extended with a sequence encoding signal peptide 176 and restriction sites for EcoRI and BamHI and was custom-synthesized by LLC “DNA-Synthesis” (Moscow, Russia).

Additionally, the integration vector was modified to include the constant region of a human IgG1 antibody [CH2-CH3] (Fc fragment), containing amino acid substitutions that enhance FcRn binding affinity [38]. Based on the pVEAL2 vector, four variants of the integration vector pVEAL2_9E2ch were generated (patent RU 2801532): pVEAL2_9E2ch-V1 (native, without substitutions), pVEAL2_9E2ch-V2 (substitutions in the H433/N434 region), pVEAL2_9E2ch-V3 (substitutions in the M252/S254/T256 region), and pVEAL2_9E2ch-V4 (containing both substitution clusters M252/S254/T256 and H433/N434).

For the production of these plasmid variants, Fc fragments incorporating the selected amino acid substitutions were custom-synthesized and cloned into the pVEAL2 vector at the AsuNHI and SalI restriction sites.

Recombinant constructs were designed using SnapGene 3.2.1.

### 4.2. Construction of the Integration Vector pMV-FcRn-B2M

The nucleotide sequence of FcRn-B2M, including the Gaussia luciferase (GL) signal peptide, the sequence corresponding to the heavy α chain of FcRn (α-FcRn), a 6× His tag for affinity purification (6His), a self-cleaving P2A peptide enabling separation of the β2-microglobulin sequence (P2A), the β2-microglobulin sequence (B2M), and restriction sites for AsuNHI and SalI, was custom-synthesized by LLC “DNA-Synthesis” (Moscow, Russia) and cloned into the pMV vector. A vector map of the recombinant construct, designed using SnapGene 3.2.1, is presented in the Appendix A (Appendix A).

### 4.3. Generation of Producer Cell Lines

Suspension cultures of CHO-K1 cells (Chinese hamster ovary, proline-auxotrophic CHO clone; FSRI SRC VB “Vector,” Rospotrebnadzor, Russia) were maintained in HyClone™ HyCell™ CHO medium (Cytiva, Marlborough, MA, USA) supplemented with 4 mM GlutaMAX (Gibco, Carlsbad, CA, USA) in 50 mL Falcon-type tubes. Cultures were incubated in a Heracell™ VIOS 160i CO_2_ Incubator (Thermo Scientific, Waltham, MA, USA) at 37 ± 1 °C, 5% CO_2_, 80% humidity, and shaken at 185 rpm on an orbital shaker (Infors-HT, Minitron, Infors AG, Bottmingen, Switzerland) until reaching a density of 6–8 × 10^6^ viable cells/mL.

Immediately prior to transfection, cell concentration and viability were assessed using a TC20 cell counter (Bio-Rad Laboratories, Hercules, CA, USA). A volume of suspension containing 5 × 10^6^ viable cells was collected, centrifuged at 800 rpm for 5 min at 4 °C (1580R, Gyrozen, Gimpo, Republic of Korea), and the supernatant was removed. The cell pellet was resuspended in 1 mL of HyClone HyCell TransFx-C transfection medium (Cytiva, Marlborough, MA, USA) and transferred to a 50 mL Falcon tube.

Transfection was performed by adding 6.75 μg of plasmid DNA pVeal2_9E2ch (variants V1–V4 or pMV-FcRn-B2M) and 0.75 μg of plasmid pSB100 (Addgene, Watertown, MA, USA) to the cell suspension, followed by 10 μg of polyethyleneimine (PEI 25K™). The mixture was incubated for 4 h at 200 rpm, 37 ± 1 °C, in a humidified atmosphere with 5% CO_2_. Afterward, 4 mL of pre-warmed HyClone™ HyCell™ CHO growth medium supplemented with 4 mM GlutaMAX was added, and cells were further cultured for 48 h under the same conditions.

High-producing monoclonal clones were subsequently isolated from the polyclonal pool using limiting dilution. Selected monoclonal producer clones were frozen in a working cell bank for subsequent experiments.

### 4.4. Suspension Cultivation of Producer Cell Lines

Selected monoclonal-producing clones were thawed from the working cell bank. The initial passage was performed in a 50 mL tube with a working volume of 5 mL of HyClone™ HyCell™ CHO medium supplemented with 4 mM GlutaMAX at a cell density of 1 × 10^6^ cells/mL. Cultures were maintained on a shaker at 200 rpm, 37 ± 1 °C, in a humidified atmosphere containing 5% CO_2_. Cell growth and viability were monitored every 48–72 h using a TC20 cell counter. Subsequent passages were performed when cell density reached 6–8 × 10^6^ cells/mL. Once the culture demonstrated stable growth and high viability, the working volume of the suspension was gradually increased in a stepwise manner.

Cells were then cultured to the target volume at a density of 6–8 × 10^6^ viable cells/mL, viability of approximately 95%, and a glucose concentration of 40 mM. Cultivation was carried out on a shaker at 200 rpm, 31 ± 1 °C, in a humidified atmosphere with 5% CO_2_ for 14 days.

Glucose levels in the culture medium were continuously monitored during cultivation at 31 ± 1 °C using a Diacont glucometer (OK Biotech, Nanjing, China). The glucose concentration was maintained at 40 mM by supplementing the medium with sterile 40% glucose solution.

### 4.5. Protein Purification

#### 4.5.1. Purification of mAb 900

Preliminary purification of the supernatant was performed by adding 10% of the volume of the liquid sorbent Ammoflok-25 (Fizlabpribor, Moscow, Russia). The resulting precipitate and cellular debris were removed by centrifugation at 12 000 rpm for 20 min at 4 °C in an Avanti J-30I high-speed centrifuge (Beckman Coulter, Brea, CA, USA). The resulting supernatant was then filtered through a 0.22 µm filtration system (Jet Biofil, Guangzhou, China). The mAb was isolated and purified by affinity chromatography on a column (Cytiva, Marlborough, MA, USA) packed with MabSelect SuRe sorbent containing recombinant protein A ligand (Cytiva, Marlborough, MA, USA) at a rate of (75 ± 5) ml of culture supernatant per 1 mL of sorbent. The clarified culture supernatant was applied to the chromatography column, pre-equilibrated with buffer A (16.2 mM Na_2_HPO_4_ × 12H_2_O, 3.8 mM NaH_2_PO_4_ × 2H_2_O, 150 mM NaCl, pH 7.4), at a flow rate of (2.2 ± 0.5) ml/min, using an Akta pure 150 medium-pressure chromatography system with Unicorn 7.6 software (Cytiva, Marlborough, MA, USA). The column was then washed again with buffer A until a constant optical density was achieved. The target protein was eluted with a linear gradient from 0 to 100% buffer B (10 mM Na_3_C_6_H_5_O_7_ × 5.5H_2_O, pH 3.0) in a volume equal to 8 column volumes (CV), followed by washing with buffer B. During the elution process, the acidic pH of the obtained fractions was immediately neutralized to pH 7.4 by adding buffer (200 mM Na_2_H_P_O_4_ × 12H_2_O, 300 mM NaCl, pH 10.8) under the control of an Anion-4100 pH meter (Infraspak-Analit NPP, Novosibirsk, Russia). After analysis by electrophoresis in a 15% sodium dodecyl sulfate-polyacrylamide gel under reducing conditions, fractions containing mAb 900 were pooled and dialyzed (SID 9652, Sigma, Ronkonkoma, NY, USA) against buffer (40.5 mM Na_2_HPO_4_ × 12H_2_O, 9.5 mM NaH_2_PO_4_ × 2H_2_O, 50 mM NaCl, pH 7.4) at 6 ± 2 °C for 20 ± 2 h. Ion-exchange chromatography was performed using Q-Sepharose (Q Fast Flow, Cytiva, Marlborough, MA, USA). The column with the sorbent was washed with base buffer (40.5 mM Na_2_HPO_4_ × 12H_2_O, 9.5 mM NaH_2_PO_4_ × 2H_2_O, 50 mM NaCl, pH (7.4 ± 0.05)) to a pH of (7.4 ± 0.05). The mobile phase flow rate was 5 mL/min. Then the target protein solution was applied. Then the sorbent was washed with five column volumes of base buffer. The target protein, not adsorbed on Q-Sepharose, began to exit immediately during the application period. The target protein solution exiting the column was collected in tubes in fractions of (3.5 ± 0.5) ml, starting from the optical density of the solution of at least 0.1 o.u., measured at a wavelength of 280 nm. Following SDS-PAGE analysis, fractions containing the target protein were pooled and dialyzed against buffer (40.5 mM Na_2_HPO_4_ × 12H_2_O, 9.5 mM NaH_2_PO_4_ × 2H_2_O, 300 mM NaCl, pH 7.4) at 6 ± 2 °C for 20 ± 2 h. After dialysis, mAb 900 was filtered through a 0.22 μm filtration system (Jet Biofil, Guangzhou, China) and stored at 6 ± 2 °C.

#### 4.5.2. Purification of Neonatal Fc Receptor (FcRn)

FcRn was purified by affinity chromatography using an IMAC Sepharose column (Cytiva, Marlborough, MA, USA; 50 mL) equilibrated with buffer containing 30 mM NaH2PO4, 0.5 M NaCl, and 20 mM imidazole (pH 7.4). The target protein was eluted using a linear gradient of imidazole from 20 to 0.5 M in the same buffer. Fractions with an optical density ≥ 0.25 a.u. were analyzed by denaturing 15% SDS-PAGE. Fractions containing FcRn were pooled and dialyzed against 50 mM NaHCO3 (pH 7.6).

### 4.6. Biolayer Interferometry (BLI)

Measurement of the binding kinetics was performed on an Octet K2 instrument (ForteBio, Fremont, CA, USA) using NTA biosensors (Cat. #18-5101). Correction for baseline drift was performed by subtracting the average of the shifts recorded by the sensor loaded with the antibody not reacting with the antigens. The data were processed using the Data Analysis HT 12.0.1.55 software. The values obtained by the control sensor were read from all other results, including the highest value of the background signals. To do this, when processing the results, a reference sensor was selected, and reference wells for subtraction were also selected. Crooked competitors were the result of dissociation. The experimental model was used in a 1:1 ratio.

#### 4.6.1. Interaction Between FcRn and Recombinant Antibody with Different Fc-Fragments

FcRn and the comparative protein Sag1 were prepared at 30 μg/mL, while mAb 900 variants (V1–V4) were diluted to 1, 2, and 4 μg/mL and added to microplate wells at 200 μL per well. Microplates with samples and biosensors were installed in the Octet K2. Sensors were pre-washed in phosphate-buffered saline (PBS) for 10 min with shaking at 1000 rpm. For ligand loading, the first sensor was immersed in a well containing FcRn, and the second sensor in a well containing Sag1, and incubated for 300 s at 1000 rpm. Unbound analytes were removed by washing sensors in PBS for 30 s at 1000 rpm. Association kinetics were measured by immersing both sensors in wells containing the respective antibody variant at 1 μg/mL for 180 s at 1000 rpm. Dissociation was monitored by transferring sensors to PBS for 320 s under continuous shaking at 1000 rpm. Cyclic regeneration was performed using 50 mM glycine-HCl (pH 1.5) and PBS wash buffer for three cycles, with sensors incubated for 10 s at 1000 rpm during each step. Subsequently, nickel was immobilized on the sensor surface by immersing the sensors in 10 mM nickel chloride for 30 s at 1000 rpm. The procedure was repeated with increasing antibody concentrations. For each antibody variant (V1–V4), three replicate measurements were conducted at three concentrations ranging from 12.8 to 51.3 nM. KDs were calculated for all FcRn–antibody pairs.

#### 4.6.2. Interaction Between D III and Antibodies mAb 9E2, mAb 900

Antibodies were prepared at concentrations of 25 μg/mL (mAb 9E2) and 18.5 μg/mL (mAb 900). Recombinant DIII protein was diluted to 20 μg/mL. Recombinant human mAb IB20 (25 μg/mL) [26], which does not bind to DIII, was included as a negative control. Sample plates and biosensors were installed in the Octet K2 system. Sensors were pre-washed in PBS for 10 min with shaking at 1000 rpm. For ligand loading, sensors were immersed in wells containing DIII for 300 s at 1000 rpm. Unbound DIII was removed by washing the sensors in PBS for 30 s at 1000 rpm. Association kinetics were measured by immersing sensors in wells containing mAb 9E2, mAb 900, or IB20 at 1 μg/mL and incubating for 180 s at 1000 rpm. Dissociation was monitored by transferring sensors to PBS for 320 s under continuous shaking at 1000 rpm. Kinetic analysis included a 60-s baseline in PBS. Cyclic sensor regeneration was performed as described in Section 4.6.1. Each antibody (mAb 9E2 and mAb 900) was analyzed in triplicate at five different concentrations ranging from 10.4 to 166.4 nM, with a twofold serial dilution between concentrations.

### 4.7. Reversed-Phase High-Performance Liquid Chromatography (RP-HPLC)

RP-HPLC was conducted on an LC-20 Prominence system (Shimadzu, Kyoto, Japan) equipped with a spectrophotometric detector, an autosampler, and a column thermostat. Separation was achieved using a Kromasil 300-5-C4 column (4.6 × 250 mm). Data acquisition and processing were performed using LabSolutions software, version 5.95 (Shimadzu, Kyoto, Japan).

The mobile phases employed were as follows: buffer A, 0.1% trifluoroacetic acid (TFA) in deionized water, and buffer B, 0.1% TFA in 70% isopropanol/water. RP-HPLC analysis was carried out under a gradient elution program.

0.0 min. → 10% B,

15.0 min → 100 B,

20.0 min → 100 B,

20.1 min → 10% B,

28.0 min → 10% B.

The mobile phase was delivered at a flow rate of 1.0 mL/min, and the injection volume was 50 μL. Column temperature was maintained at 80 °C throughout the analysis. Chromatograms were monitored at a wavelength of 214 nm.

### 4.8. ELISA

ELISA was performed using a standard protocol with sequential twofold serial dilutions of antibodies, each tested in triplicate. Ninety-six-well Nunc MaxiSorp polystyrene plates (Thermo Scientific, Waltham, MA, USA) were coated either with a recombinant DIII domain of the WNV envelope glycoprotein at 150 ng/well in 10 mM carbonate buffer (pH 9.5) at 4 °C for 20 h, or with WNV, LEIV-Vlg99-27889-human at 200 ng/well. Non-specific binding sites were blocked with 0.2% casein in phosphate-buffered saline (PBS, pH 7.4) for 1 h at 37 °C. Test antibodies were serially diluted in PBS to concentrations from 500 μg/mL to 0.020 μg/mL. Murine mAb 29F10 targeting the NS1 protein of Tick-borne encephalitis virus (TBEV, I-1022, Biosan, Novosibirsk, Russia) and recombinant human antibody IB20 were employed as negative controls. Chromogenic detection was carried out using rabbit anti-mouse and anti-human immunoglobulin antibodies conjugated to horseradish peroxidase (anti-Mouse-HRP, anti-Human-HRP, Imtek, Moscow, Russia) at concentrations recommended by the manufacturer. Tetramethylbenzidine (TMB) substrate (Tetramethylbenzidine Liquid Substrate System for ELISA, Sigma-Aldrich, Burlington, MA, USA) was used for color development, and the reaction was terminated by adding 0.5 M sulfuric acid. Optical density (OD) was measured at 450 nm using a Thermo Scientific Varioskan LUX spectrophotometer (Thermo Scientific, Waltham, MA, USA).

The endpoint dilution was defined as the highest dilution yielding an OD greater than the critical value (OD_crit), calculated as follows: OD_crit = mean OD of the sample at 1:100 dilution + 0.15.

### 4.9. Study of Acute Toxicity of mAb 900 Following Intraperitoneal Administration in Mice

The study used a mAb 900 at a concentration of 1.1 mg of protein/mL. The control group received sterile 0.9% sodium chloride solution.

The study was conducted on 40 outbred ICR mice (20 males and 20 females) weighing 18–20 g. The animals were housed under standard vivarium conditions (temperature 20–24 °C, humidity 45–65%, with ad libitum access to water and standard pelleted diet) in accordance with GOST 33216-2014. A 48-h acclimatization period was observed prior to the experiment. Using a random sampling method, mice were balanced by body weight and divided into two groups of 20 animals each (10 males + 10 females). Experimental group, single intraperitoneal (i.p.) administration of mAb 900 at a dose of 550 µg per animal (injection volume 0.5 mL). Control group, single i.p. administration of an equivalent volume (0.5 mL) of physiological saline.

The selected dose corresponded to one proposed human dose, recalculated for mice based on body surface area.

Animals were observed for 7 days post-injection. General condition, behavioral responses, feed and water consumption, and body weight were recorded daily.

Laboratory and instrumental investigations on days 1 and 7 included hematological analysis, post-mortem (pathological) examination, and histological examination.

Blood for hematological analysis was collected from the retro-orbital sinus. Standard parameters (white blood cells, red blood cells, platelets, hemoglobin, hematocrit) were determined using an automatic hematology analyzer MicroCC-20 Plus VET (High Technology, North Attleboro, MA, USA), with the leukocyte formula counted manually.

Following euthanasia, a necropsy was performed. Internal organs were assessed macroscopically, weighed, and their relative organ weight indices were calculated (organ weight, mg/body weight, g × 10).

For histological examination, samples from 14 key organs and tissues were collected: brain, thymus, lungs, heart, liver, spleen, kidneys, adrenal glands, gastrointestinal tract, bone marrow, reproductive organs, injection site, and regional lymph nodes. The material was fixed in 10% neutral buffered formalin and processed according to a standard histological protocol: paraffin embedding, preparation of 4–5 µm thick sections, and staining with hematoxylin and eosin. Evaluation was performed by light microscopy using an Axio Imager Z1 microscope (Carl Zeiss, Göttingen, Germany), with particular attention paid to signs of inflammation, degeneration, dystrophy, and microcirculation disorders.

The obtained data were processed using the Statgraphics 5.0 software package (Statgraphics Technologies, Inc., The Plains, VA, USA). The Shapiro–Wilk test was applied to check for normal distribution. Non-parametric methods were used to assess intergroup differences: the Kruskal–Wallis H-test followed by pairwise comparison using the Mann–Whitney U-test. Differences were considered statistically significant at *p* < 0.05.

### 4.10. Viruses, Cell Cultures and Neutralization Assay

The study experiments used the WNV strain WNV, LEIV-Vlg99-27889-human, obtained from the State Collection of Pathogenic Microorganisms and Rickettsiae of the State Research Center of Virology and Biotechnology «Vector». The virus was cultured on Vero monolayer cell cultures, grown to 80–90% confluency in DMEM F12 medium (Gibco, USA) containing 10% fetal bovine serum (Gibco, Carlsbad, CA, USA), penicillin 100 IU/mL, and streptomycin 100 μg/mL (Gibco, Carlsbad, CA, USA) in an atmosphere with 5% CO_2_ at 37 °C. The infectious activity of the viral substances was determined by microtitration on 96-well culture plates (Greiner AG, Kremsmünster, Austria) with a subconfluent monolayer of cells, as previously described [39]. The calculation of viral infectious titers was performed using the Spearman-Karber method and expressed as log10 TCID_50_ (50% tissue culture infectious dose). The viral substances were stored at a temperature of −80 °C

WNV neutralization was performed using a microneutralization assay on Vero cell cultures (96-well culture plates, 5 × 10^4^ cells/well). Both the virus and antibodies were diluted in DMEM F12 medium. A mixture of virus (100 μL) and threefold serial dilutions of antibodies (100 μL) was incubated for two hours at room temperature, after which the mixtures were used to infect Vero cell monolayers (4 repetitions per experimental point infected with 100 TCID_50_ WNV). Results were assessed on day 7 post-infection by microscopic scoring of the cytopathic effects (CPE) and by measuring cell viability in the formazan-based MTT assay described previously [39]. Antibody titers were defined as the highest dilution that protected 50% of the cell monolayers from virus-induced CPE. SARS-CoV-2 RBD-specific human mAb iB20 was used as negative control.

### 4.11. Determination of the Thermal Stability of mAb 900

The study was conducted using CD spectroscopy on a Chirascan VX-100 spectropolarimeter (Applied Photophysics, Leatherhead, UK). Measurements were performed in a cuvette with a 1 mm optical path length. A 200 µL sample of mAb 900 was placed in the cuvette. Spectra were recorded in the temperature range of 5 to 100 °C with a 5 °C step. The temperature was verified using a thermocouple placed directly in the cuvette. The wavelength range was 200–280 nm with a 1 nm step, a signal integration time of 0.5 s per wavelength, and a slit width of 1 nm.

The baseline was constructed using the buffer in which mAb 900 was dissolved (40.5 mM Na_2_HPO_4_·12H_2_O, 9.5 mM NaH_2_PO_4_·2H_2_O, 300 mM NaCl, pH 7.4).

The acquired data were analyzed using specialized Global Thermal Analysis 3 software (Applied Photophysics, Leatherhead, UK), which enables the simultaneous analysis of a set of CD spectra obtained at different temperatures. The theoretical model considered was that of intramolecular protein denaturation. Based on the spectral data, the melting temperature (Tm) and the enthalpy of denaturation (ΔH) for mAb 900 were determined.

### 4.12. Size-Exclusion Chromatography (SEC)

Native-state SEC was performed by gel filtration using a 10 mL Sephacryl S-200 HR column on an Acta Pure 150 chromatography system (Cytiva, Marlborough, MA, USA). The separation was carried out in PBS, pH 7.4, at a flow rate of 0.5 mL/min. The column temperature was maintained at 2–8 °C. The column was calibrated with proteins of known molecular masses: lysozyme (14.3 kDa), β-lactoglobulin (18.4 kDa), egg albumin (45 kDa), bovine serum albumin (66 kDa), and γ-globulin (150 kDa). A calibration curve was constructed from these data and used to determine the molecular weight of the target protein.

### 4.13. Glycosylation Profile Analysis of mAb 900

The glycosylation profile was analyzed for two mAb 900 samples (sample 1 and sample 2) produced under identical conditions with a two-month interval. Samples 1 and 2 were subjected to enzymatic release of N-glycans from the protein backbone using PNGase F (New England Biolabs, Ipswich, MA, USA). The released glycans were labeled with the fluorescent tag 2-aminobenzamide (2-AB). The analysis was performed by HILIC-HPLC with fluorometric detection in gradient mode on an UltiMate 3000 system (Thermo Scientific, Sunnyvale, CA, USA).

All glycan peaks were integrated on the chromatograms, identifying eight major peaks corresponding to G0-N, G0F-N, G0, G0F, Man5, G1, G1F, and G2F glycans. The G1 and G1F glycans represent a set of two isomers each, eluting as two adjacent peaks.

Using Chromeleon 7.2 CDS software, the relative content of each glycan (Bi, %) was calculated. Subsequently, the percentage of afucosylated glycans (AF, %) and galactosylated glycans (G, %) were determined using the following formulas:AF = BG0-N + BG0 + BMan5 + BG1,(1)
where BG0-N, BG0, BMan5, BG1 is the relative content of the corresponding glycan, %.G = BG1 + BG1F + 2 × BG2F,(2)
where BG1, BG1F, BG2F is the relative content of the corresponding glycan, %.

### 4.14. Assessment of the Protective Efficacy of mAb 900 In Vivo

The protective activity of mAb 900 was evaluated in female BALB/c mice (weight 8–11 g), with six mice per group. The mice were obtained from the breeding facilities of laboratory animals of the State Research Center of Virology and Biotechnology “Vector”. All animal maintenance procedures complied with the European Directive 2010/63/EU on the protection of animals used for scientific purposes. The study was approved by the Bioethics Committee of the State Research Center of Virology and Biotechnology “Vector” (Protocol No. 4, dated 11 April 2025; Registration number: State Research Center of Virology and Biotechnology “Vector”/06-04.2025).

For infection, WNV strain LEIV-Vlg99-27889-human was used. Mice were infected via intraperitoneal (i.p.) injection with a dose of 0.3 × 10^6^ LD_50_ per mouse in a volume of 300 µL. mAb 900 was administered i.p. at 6 h post-infection in a volume of 300 µL per mouse. The following therapeutic doses of mAb 900 were tested for protective efficacy: 10 µg, 50 µg, 100 µg, and 200 µg per animal.

A virus control group was established by infecting mice with the selected dose of WNV without subsequent mAb 900 therapy. A safety control for the antibody was performed by administering mAb 900 at a dose of 200 µg per mouse to uninfected animals.

### 4.15. Statistical Analysis

Statistical data processing was conducted using the statistical program STATISTICA 12 (StatSoft Inc., Tulsa, OK, USA), Excel (Microsoft Corp., Redmond, WA, USA), GraphPad Prism 8.2.1 (GraphPad Software, Inc., San Diego, CA, USA). Statistical evaluation of the differences between the groups was performed using Student’s *t*-test; *p* < 0.05 was considered significant.

## 5. Conclusions

In this study, a recombinant chimeric virus-neutralizing mAb 900 targeting WNV was successfully generated. Our work demonstrates that the chimerization and rational Fc mutagenesis (M252Y/S254T/T256E and H433K/N434F) had no effect on its antiviral function.

The chimeric antibody retained high-affinity binding to the WNV DIII domain and, most importantly, fully conserved the neutralizing activity of its murine progenitor *in vitro*. The introduced Fc modifications endowed mAb 900 with significantly enhanced affinity for the neonatal Fc receptor, a direct predictor of an extended serum half-life. Beyond its functional integrity, mAb 900 exhibited high conformational stability and a favorable biophysical profile suitable for development.

The most significant advance presented here is the demonstration of *in vivo* protective efficacy. In a lethal challenge model using a neuroinvasive WNV strain, a single, post-exposure dose of mAb 900 (100 µg) conferred significant protection, with 83.3% survival.

Collectively, these results establish mAb 900 as a promising preclinical candidate against WNV.

## 6. Patents

The results concerning the plasmid genetic construct pVeal2_9E2ch and cell line CHO-K1-900 are also presented in the patent RU 2 801 532 C1 “pVEAL2-9E2ch-scFv plasmid genetic construct, strain of recombinant cell line CHO-K1-9E2ch and chimeric single-chain antibody 9E2ch against West Nile virus produced by the specified strain of cell line CHO-K1-9E2ch, with high affinity for neonatal FcRn receptor”.

## Figures and Tables

**Figure 1 ijms-26-12181-f001:**
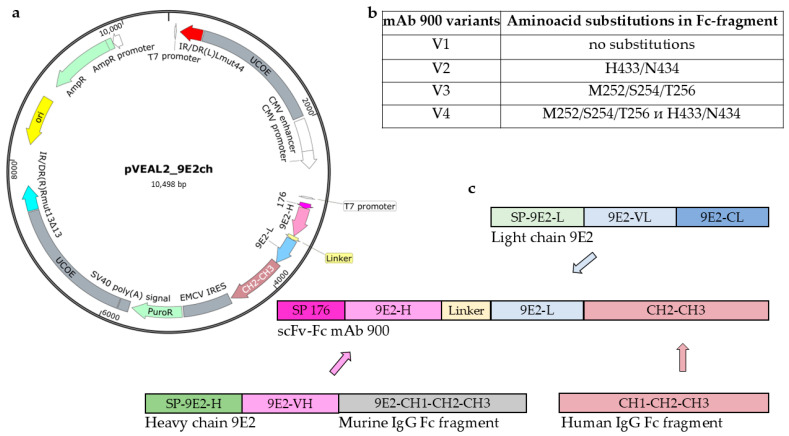
Structure of expression plasmids. (**a**) pVEAL2_9E2ch map. The plasmid contains the following elements: ori—the origin of replication, the IR/DR—direct and inverted repeat sequences containing SB100 transposase binding sites, the UCOE sequences preventing chromosomal silencing, the CMV enhancer, the CMV promoter, the signal peptide 176 (SP 176)—a sequence encoding a hybrid signal peptide of luciferase and fibroin that ensures protein export from the cell, the 9E2-H, 9E2-L sequences encoding variable fragments of the heavy and light chains of the monoclonal antibody (mAb) 9E2 connected by a peptide linker, the CH2-CH3—fragment of the constant chain of the human IgG1 antibody, the EMCV IRES—internal ribosome entry site, the *PuroR*—puromycin antibiotic resistance gene, the SV40 poly(A) signal; ampicillin resistance gene *AmpR*; (**b**) variants of amino acid substitutions in the Fc fragment of mAb 900. Variant V1 does not contain amino acid substitutions in the human IgG Fc fragment. Variant V2 contains amino acid substitutions H433/N434 in the human IgG Fc fragment. Variant V3 contains amino acid substitutions M252/S254/T256 in the human IgG Fc fragment. Variant V4 contains amino acid substitutions H433/N434 and M252/S254/T256 in the human IgG Fc fragment; (**c**) Scheme of expression plasmid construction.

**Figure 2 ijms-26-12181-f002:**
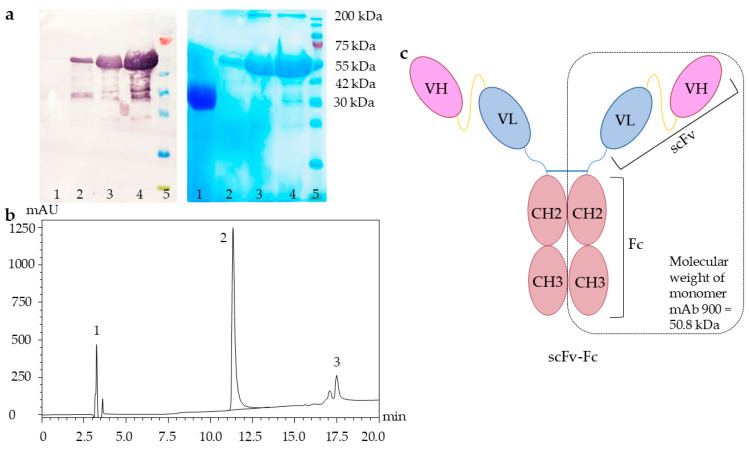
Analysis of purified recombinant mAb 900. (**a**) Western blot (**left panel**) and SDS-PAGE (**right panel**) analysis. Lane 1: negative control (SARS-CoV-2 RBD); lanes 2–4: mAb 900 (1, 5, and 10 µL, respectively); lane 5: prestained protein molecular weight marker (10–200 kDa). (**b**) Reversed-phase high-performance liquid chromatography (RP-HPLC) profile. Peak 1: low-molecular-weight impurities; peak 2: monomeric mAb 900; peak 3: high-molecular-weight aggregates. (**c**) Schematic representation of the scFv-Fc format of mAb 900.

**Figure 3 ijms-26-12181-f003:**
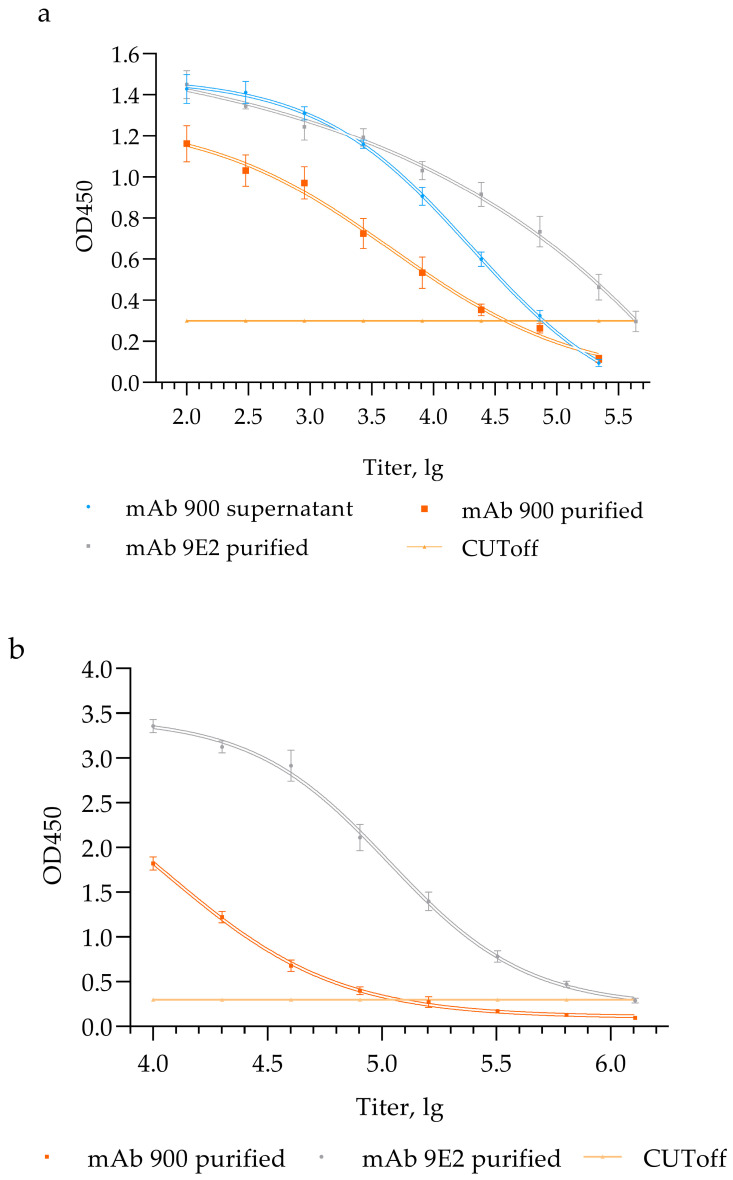
Functional activity of chimeric mAb 900 and murine mAb 9E2 in Enzyme-Linked Immunosorbent Assay (ELISA). (**a**) Interaction of mAb 900 supernatant, purified mAb 900 and purified mAb 9E2 with inactivated WNV, LEIV-Vlg99-27889-human; (**b**) interaction of purified mAb 900 and purified mAb 9E2 with a recombinant analog of D III of WNV glycoprotein E. ELISA optical density values represent means ± SD of three independent experiments.

**Figure 4 ijms-26-12181-f004:**
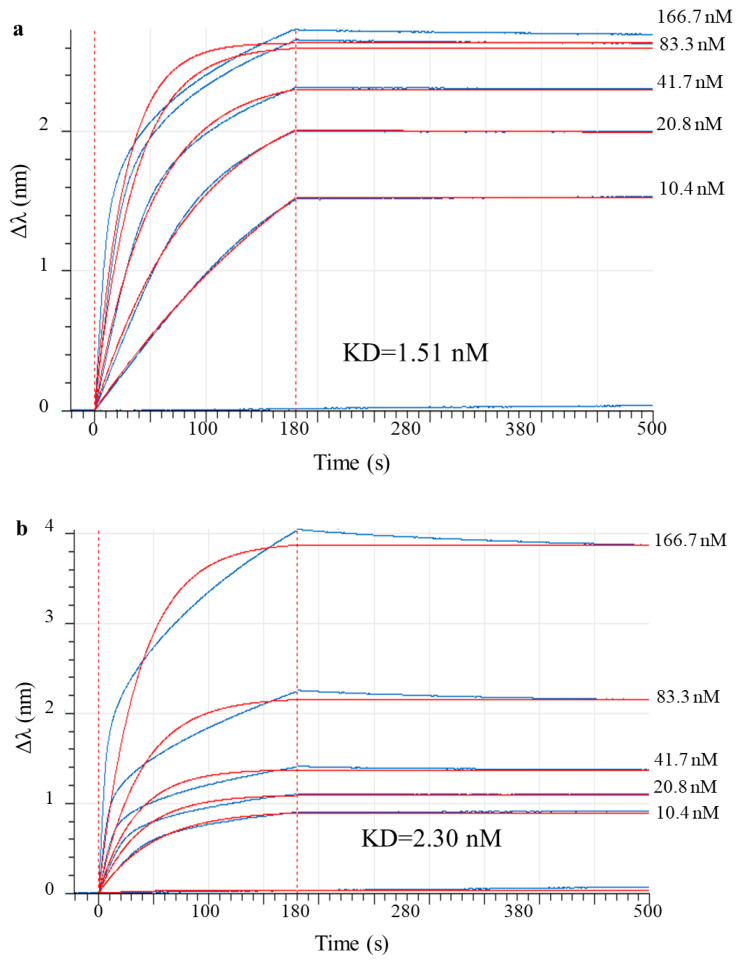
Sensorgram curves for recombinant analog of D III of WNV glycoprotein E binding of increasing concentrations of murine mAb 9E2 and chimeric mAb 900 in a biolayer interferometry (BLI)-kinetic assay. The blue line indicates the values obtained from the interaction of recombinant antigen with mAb 9E2 and mAb 900 at different concentrations (166.7 nM, 83.3 nM, 41.7 nM, 20.8 nM, 10.4 nM); the red line represents the global fit of the 1:1 binding model, approximating the experimental data. (**a**) Sensorgram curves for DIII binding of mAb 9E2; (**b**) Sensorgram curves for DIII binding of mAb 900.

**Figure 5 ijms-26-12181-f005:**
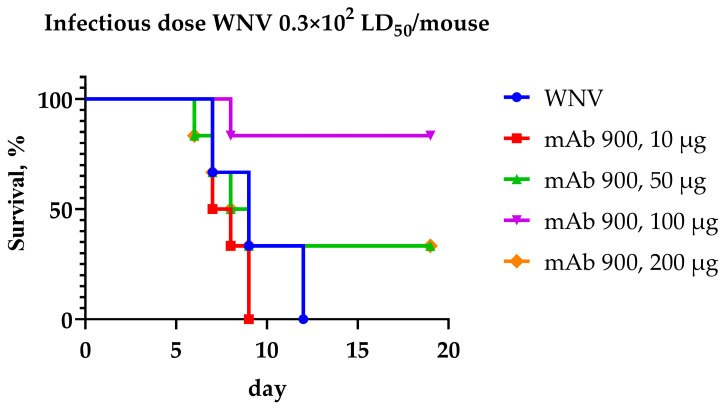
Survival curves of mice treated with various doses of mAb 900 following infection with the WNV strain LEIV-Vlg99-27889-human (culture) at a dose of 0.3 × 10^2^ LD_50_ per mouse.

**Table 1 ijms-26-12181-t001:** Equilibrium dissociation constants (KD) values of mAb 900–neonatal Fc receptor (FcRn) interactions.

mAb 900 Variants	KD, M	KD Error, M
V1 ^1^	4.35 × 10^−6^	7.81 × 10^−4^
V2	2.92 × 10^−9^	1.09 × 10^−10^
V3	2.81 × 10^−8^	2.00 × 10^−8^
V4	1.41 × 10^−9^	1.57 × 10^−11^

^1^ For variant V1, the KD values were not reproducible over three independent measurements. The analysis of this variant was affected by a high level of noise, leading to substantial measurement variability.

**Table 2 ijms-26-12181-t002:** ELISA mAb 900 and mAb 9E2 with inactivated WNV, LEIV-Vlg99-27889-human.

mAb	Concentration (mg/mL)	Titer, lg	Normalized Titer, lg
mAb 900 supernatant	1.5	4.38 ± 0.38	4.20
mAb 900 purified	0.2	3.90 ± 0.29	4.61
mAb 9E2 purified	5	5.34 ± 0.40	4.64

Notes: ELISA titer values represent means ± SD of three independent experiments. SARS-CoV-2 RBD-specific human mAb iB20 (Titer—<1.0 lg) and murine mAb 29F10 targeting the NS1 protein of Tick-borne encephalitis virus (Titer—<1.0 lg) were employed as negative controls. Student’s *t*-test was used for two-group comparisons.

**Table 3 ijms-26-12181-t003:** The results of the neutralization assay of mAb 900 and mAb 9E2 against 100 TCID_50_ WNV.

mAb	Concentration (mg/mL)	Titer, lg	IC_50_ (ng/mL)
mAb 900 supernatant	1.5	3.91 ± 0.27	185 ± 38
mAb 900 purified	0.2	3.43 ± 0.24	74 ± 16.3
mAb 9E2 purified	5	4.86 ± 0.33	68.6 ± 13.8
mAb iB20 purified	0.5	<1.0	>50,000

Notes: neutralization titer and IC_50_ values represent means ± SD of three independent experiments. SARS-CoV-2 RBD-specific human mAb iB20 [26] was used as negative control. Student’s *t*-test was used for two-group comparisons.

**Table 4 ijms-26-12181-t004:** Protective efficacy of recombinant chimeric mAb 900 against WNV (LEIV-Vlg99-27889-human) infection at a dose of 0.3 × 10^2^ LD_50_ per mouse. The ratio of surviving (s) to total (t) animals is indicated in parentheses (s/t).

mAb 900 (200 μg)	mAb 900 (100 μg)	mAb 900 (50 μg)	mAb 900 (10 μg)	WNV (Virus Control)	mAb 900 (mAb Control) 200 μg/Mouse
33% (2/6)	83.3% (5/6)	33% (2/6)	0 (0/6)	0 (0/6)	100% (6/6)

## Data Availability

The original contributions presented in this study are included in the article/Appendix A. Further inquiries can be directed to the corresponding author.

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
