# Peer review of "Does the Chimerization Process Affect the Immunochemical Properties of WNV-Neutralizing Antibody 900?"

_ijms, 2025, doi:10.3390/ijms262412181_

Round 1

Reviewer 1 Report

Comments and Suggestions for Authors

I have reviewed the manuscript “Does the chimerization process affect the immunochemical properties of WNV-neutralizing antibody 900?” submitted by Anastasiya A. Isaeva et al.

In this paper the authors described their modification of the mouse monoclonal antibody 9E2, which was previously shown to neutralize West Nile Virus (WNV). They fused the IgV domain of the mouse mAb with a modified version of human IgC in order to optimize the interaction of the modified antibody construct with the neonatal Fc receptor (FcRn), and determined KD values between different modifications of the Fc part of the antibody and the FcRn. Thereafter, they compared the original mouse antibody 9E2 and the modified antibody construct 900 in terms of their activity in antigen recognition, virus neutralization, and affinity to the recombinant DIII protein of the virus surface. The authors found that the modification of the mAb did not or only slightly influence functional performance, while the modification of the Fc part significantly enhance the binding to the FcRn, indicating that the analyzed antibody construct warrens in vivo evaluation.

This is a valid investigation, which is well written, nevertheless some points need further attention of the authors, in order to improve the manuscript.

Figure 1. The figure legend should be extended, e.g. a. describe the highlight of the plasmid, SP176 should be spelt out, the modification mentioned in b. could be explained.

Figure 2. Figure legend: Title should be “Analyses of recombinant antibodies” a. left picture insert “western blot” and right picture “SDS page”. Lane1, negative control lane 2,3,4, mAb ….. similarly for b. and c.

Figure 3. include the name of the antibody into the figure 3a and 3b. and expand a bit the fig. legend.

Page 7. 2.5. Title: explain MAT 900

Figure 4. legend: the brown line indicates the average value – average of what? I did not see a brown line.

Author Response

Summary

The team of authors would like to thank the reviewer for high evaluation of our work and making important comment.

Point-by-point response to Comments and Suggestions for Authors

Comments 1: Figure 1. The figure legend should be extended, e.g. a. describe the highlight of the plasmid, SP176 should be spelt out, the modification mentioned in b. could be explained.

Response 1: Thank you for your comment. We have added more information about plasmid in the figure caption.

Page 3, paragraph 2.1. Fc Fragment Engineering of mAb 900, lines 107-121. Figure 1. Structure of expression plasmids. (a) pVEAL2_9E2ch map. The plasmid contains the following elements: ori − the origin of replication, the IR/DR direct and inverted repeat sequences containing SB100 transposase binding sites, the UCOE sequences preventing chromosomal silencing, the CMV enhancer, the CMV promoter, the signal peptide 176 (SP 176) - a sequence encoding a hybrid signal peptide of luciferase and fibroin that ensures protein export from the cell, the 9E2-H, 9E2-L sequences encoding variable fragments of the heavy and light chains of the 9E2 antibody connected by a peptide linker, the CH2-CH3 fragment of the constant chain of the human IgG1 antibody, the EMCV IRES internal ribosome entry site, the PuroR puromycin antibiotic resistance gene, the SV40 poly(A) signal; ampicillin resistance gene AmpR; (b) Variants of amino acid substitutions in the Fc fragment of mAb 900. Variant V1 does not contain amino acid substitutions in the human IgG Fc fragment. Variant V2 contains amino acid substitutions H433/N434 in the human IgG Fc fragment. Variant V3 contains amino acid substitutions M252/S254/T256 in the human IgG Fc fragment. Variant V4 contains amino acid substitutions H433/N434 and M252/S254/T256 in the human IgG Fc fragment; (c) Scheme of expression plasmid construction.

Comments 2: Figure 2. Figure legend: Title should be “Analyses of recombinant antibodies” a. left picture insert “western blot” and right picture “SDS page”. Lane1, negative control lane 2,3,4, mAb ….. similarly for b. and c.

Response 2: Thank you for your comment. We titled the figure "Analysis of purified mAb 900" because it presents data only for recombinant mAb 900. We have corrected the figure legend according to the comments.

Page 5, paragraph 2.2. Cultivation of the CHO-K1-900 Cell Line and Purification of Chimeric mAb 900, lines 156-161. Figure 2. Analysis of purified recombinant mAb 900. (a) Western blot (left panel) and SDS-PAGE (right panel) analysis. Lane 1: negative control (SARS-CoV-2 RBD); lanes 2–4: mAb 900 (1, 5, and 10 µl, respectively); lane 5: prestained protein molecular weight marker (10–200 kDa). (b) Reversed-phase high-performance liquid chromatography (RP-HPLC) profile. Peak 1: low-molecular-weight impurities; peak 2: monomeric mAb 900; peak 3: high-molecular-weight aggregates. (c) Schematic representation of the scFv-Fc format of mAb 900.

Comments 3: Figure 3. include the name of the antibody into the figure 3a and 3b. and expand a bit the fig. legend.

Response 3: Thank you for your comment. Each curve on the graph has a caption below it. We chose not to label the curves on the graph itself to avoid cluttering the figure. The fig. legend was made more detailed.

Page 6, paragraph 2.3. Functional Activity of mAb 900 in ELISA, lines 190-194. Figure 3. Functional activity of chimeric mAb 900 and murine mAb 9Е2 in Enzyme-Linked Im-munosorbent Assay (ELISA). (а) Interaction of mAb 900 supernatant, purified mAb 900 and puri-fied mAb 9Е2 with inactivated WNV, LEIV-Vlg99-27889-human; (b) Interaction of purified mAb 900 and purified mAb 9Е2 with a recombinant analogue of D III of WNV glycoprotein E. ELISA optical density values represent means ± SD of three independent experiments.

Comments 4: Page 7. 2.5. Title: explain MAT 900

Response 4: Thank you for pointing this out. This title was incorrectly translated into English. We've corrected the translation.

Page 8, line 234. 2.5 KD Estimate for mАb 900.

Comments 5: Figure 4. legend: the brown line indicates the average value – average of what? I did not see a brown line.

Response 5: Thank you for pointing this out. Indeed, the figure legend is incorrect. We corrected it as follows:

Page 8, paragraph 2.5 KD Estimate for mАb 900. lines 239-245. Figure 4. Sensorgram curves for recombinant analogue of D III of WNV glycoprotein E binding of increasing concentrations of murine mAb 9E2 and chimeric mAb 900 in a BLI-kinetic assay. The blue line indicates the values obtained from the interaction of recombinant antigen with mAb 9E2 and mAb 900 at different concentrations (166.7 nM, 83.3 nM, 41.7 nM, 20.8 nM, 10.4 nM); the red line represents the global fit of the 1:1 binding model, approximating the experimental data. (a) Sensorgram curves for DIII binding of mAb 9E2; (b) Sensorgram curves for DIII binding of mAb 900.

Reviewer 2 Report

Comments and Suggestions for Authors

The authors described the generation of a chimeric scFv-Fc monoclonal antibody (mAb 900) derived from the murine anti-WNV antibody 9E2 and engineered with two Fc mutation clusters intended to enhance FcRn binding. The authors compared four Fc variants, select the highest-affinity construct (V4), purified the corresponding antibody from CHO cells, and evaluated its antigen binding, FcRn affinity, and virus-neutralizing activity through ELISA, BLI, and microneutralization assays. The study concluded that Fc engineering and chimerization preserved the immunochemical properties of the parental murine antibody and improved FcRn affinity, suggesting therapeutic potential. While the study concept is relevant, the manuscript has several major concerns, insufficient experimental rigor, and overstated conclusions. The following are the major concerns;

- The engineering strategy presented lacks substantial innovation; scFv-Fc constructs and the specific FcRn-enhancing mutations used here (M252/S254/T256 and H433/N434) are well known and widely implemented. The manuscript does not clearly articulate how this chimeric antibody advances the field beyond established methods or previously published data on the parental 9E2 antibody.

- Essential analyses are missing, including in vivo efficacy studies, pharmacokinetic assessment, Fc effector function evaluation (ADCC, CDC, FcγR binding), and glycosylation profiling. These omissions critically limit the ability to support claims of therapeutic potential or improved half-life. The manuscript focuses almost exclusively on FcRn affinity and basic binding assays, which alone are insufficient for evaluating an antiviral therapeutic candidate.

- Both SDS-PAGE and RP-HPLC indicate the presence of aggregates, yet no quantification, stability analysis, or aggregation-removal strategy (e.g., SEC) is presented. The reported molecular weight of ~50 kDa contradicts expected scFv-Fc dimer mass. No thermal stability, epitope mapping, or structural validation is provided, leaving uncertainty regarding the integrity of the engineered antibody.

- The neutralization IC50 values for mAb 900 (74 µg/mL) and the murine 9E2 control (68.6 µg/mL) are considerably higher (weaker) than what is typically required for therapeutic antiviral antibodies. This limitation is not adequately acknowledged or discussed and raises concerns about the feasibility of clinical translation.

- The BLI assays use few antibody concentrations and display noise and variability that undermine the reliability of KD calculations. ELISA results are presented as endpoint titers without binding curves, statistical analysis, or quantitative comparisons. Neutralization assays lack adequate controls, replicate numbers, and detailed reporting of infection conditions (e.g., MOI). Collectively, these issues reduce confidence in the data.

- The manuscript makes claims about preserved functionality and therapeutic promise that are not sufficiently supported by the limited dataset. Without in vivo validation, comprehensive Fc functionality testing, or pharmacokinetic studies, statements about therapeutic viability and extended half-life remain speculative.

- The manuscript makes claims about preserved functionality and therapeutic promise that are not sufficiently supported by the limited dataset. Without in vivo validation, comprehensive Fc functionality testing, or pharmacokinetic studies, statements about therapeutic viability and extended half-life remain speculative.

- Statistical methods are largely absent, aside from a brief mention of the Dunnett test. No discussion is provided on assay reproducibility or error reporting. Purity and yield claims are not quantitatively substantiated. Essential experimental controls are not described or shown. 

Author Response

Response to Reviewer 2 Comments

Summary

3. Point-by-point response to Comments and Suggestions for Authors

Comments 1: The engineering strategy presented lacks substantial innovation; scFv-Fc constructs and the specific FcRn-enhancing mutations used here (M252/S254/T256 and H433/N434) are well known and widely implemented. The manuscript does not clearly articulate how this chimeric antibody advances the field beyond established methods or previously published data on the parental 9E2 antibody.

Response 1: We appreciate the reviewer's comment. While we agree that the individual components of the engineering strategy (scFv-Fc format, FcRn mutations) are established, the core novelty and value of this work lie not in the method itself, but in its application to a antibody 9E2 and the empirical demonstration of its functional outcome.

The central scientific question we address is whether the chimerization and Fc-modification of this particular anti-WNV antibody preserves its crucial neutralizing function. As noted in the manuscript, even conservative changes like constant domain substitution can alter interdomain interactions and affect binding. Therefore, a comprehensive immunochemical characterization was essential. Our data provide the first evidence that the functional integrity of antibody 9E2 is fully maintained after chimerization and Fc-engineering. This successful engineering creates a stable, humanized candidate with with neutralizing properties, advancing development of 9E2 as a promising candidate for further preclinical development against WNV infection.

The following fragments have been added to the text of the article:

Page 2, Introduction, lines 74-82.

Pages 10-11, 3. Discussion, lines 300-320.

In addition, in response to your comment, we have significantly expanded the experimental scope to address this very point. The novelty now lies not merely in the engineering, but in providing a comprehensive preclinical characterization package for mAb 900 that was entirely lacking for the parental 9E2 antibody.

Specifically, we have added critical data that directly transitions mAb 900 from a conceptual construct to a promising candidate for further preclinical development:

Demonstration of in vivo efficacy: The key functional advance is the first proof that mAb 900 provides significant protection (83.3% survival) in a lethal mouse model of WNV infection.

We now provide essential developability data, including:

Biophysical Stability. A high thermal melting point (~77°C) from CD spectroscopy, indicating a stable, well-folded molecule.

Aggregation State. Quantitative SEC analysis confirming a predominantly monomeric state (>98%), critical for manufacturing and safety.

Safety. Acute toxicity study showing 100% survival and no adverse effects at the highest tested dose (200 µg/mouse), establishing an initial safety window.

Glycosylation Profile. Analysis showing a typical, consistent human-like glycan pattern, confirming proper production and predicting predictable pharmacokinetics and effector functions.

Therefore, this work advances the field by delivering a holistically characterized candidate: a humanized, Fc-engineered antibody derived from 9E2, with confirmed in vitro neutralizing activity, in vivo protective efficacy, and a robust developability profile. This integrated dataset provides a solid foundation for its further development and clearly differentiates it from the previously described, functionally unvalidated parental antibody.

Comments 2: Essential analyses are missing, including in vivo efficacy studies, pharmacokinetic assessment, Fc effector function evaluation (ADCC, CDC, FcγR binding), and glycosylation profiling. These omissions critically limit the ability to support claims of therapeutic potential or improved half-life. The manuscript focuses almost exclusively on FcRn affinity and basic binding assays, which alone are insufficient for evaluating an antiviral therapeutic candidate.

Response 2:

We thank the reviewer for raising these critical points regarding the comprehensive characterization of a therapeutic antibody candidate. We fully agree that in vivo efficacy studies, detailed pharmacokinetic (PK) analysis, assessment of Fc effector functions, and glycosylation profiling are essential components of the full preclinical development pathway for any therapeutic antibody.

However, the primary objective of the present manuscript is more focused. It aims to address a fundamental and prior uncertainty: whether the chimerization process and the introduction of well-described FcRn-modulating mutations compromise the core antigen-binding and virus-neutralizing function of the parental antibody 9E2.

Thus, this study is deliberately designed as a foundational, in vitro characterization step. Its central question is not 'Is this a ready therapeutic?' but 'Does the engineering process itself destroy the key property we need?' It was imperative to confirm that the engineered molecule retains its primary antiviral activity.

In this context, we prioritized the following analyses, which we believe are necessary for the stated goals:

1. Binding affinity (BLI/ELISA). To confirm no loss of target engagement.

2. In vitro neutralization. To demonstrate preservation of the critical biological function.

3. FcRn affinity. To provide direct mechanistic evidence that the introduced mutations (M252Y/S254T/T256E, H433K/N434F) function as intended at the molecular level, forming the rational basis for the hypothesis of improved half-life.

At the same time, considering your comments, we have now clarified this focused scope more explicitly in the introduction and discussion to avoid overstatement of our claims.

Also, taking into account your comments regarding several properties important for a therapeutic antibody, we have supplemented the text of the article with additional results from our experiments, including glycosylation profiling and acute toxicity analysis.

Pages 9-10, paragraphs 2.6. Glycosylation Profile of mAb 900, 2.7. Assessment of Acute Toxicity in Mice, 2.8. Protective Efficacy of mAb 900 In Vivo, lines 248-285.

Page 11, paragraph 3. Discussion, lines 331-340.

Page 15-16, paragraph 4.9. Study of acute toxicity of mAb 900 following intraperitoneal administration in mice, lines 531-568.

Page 17, paragraphs 4.13. Glycosylation Profile Analysis of mAb 900, 4.14. Assessment of the Protective Efficacy of mAb 900 In Vivo, lines 614-649.

Supplementary material S4.

Comments 3: Both SDS-PAGE and RP-HPLC indicate the presence of aggregates, yet no quantification, stability analysis, or aggregation-removal strategy (e.g., SEC) is presented. The reported molecular weight of ~50 kDa contradicts expected scFv-Fc dimer mass. No thermal stability, epitope mapping, or structural validation is provided, leaving uncertainty regarding the integrity of the engineered antibody.

Response 3:

Thank you for your thorough and constructive critique regarding the biophysical characterization of mAb 900. Your comments are highly valuable and have allowed us to significantly improve the rigor and clarity of our manuscript. Below, we provide a point-by-point response and detail the corresponding revisions.

1. Regarding aggregates, SEC, stability, and the molecular weight observation.

Molecular Weight Clarification. We apologize for the lack of clarity in the initial submission. The SDS-PAGE analysis was indeed performed under reducing conditions, which explains the observed band at approximately 50 kDa. This corresponds to the expected molecular weight of the single polypeptide chain of the reduced scFv-Fc monomer. To avoid confusion, we have now explicitly stated the electrophoresis conditions in the revised figure legend and the Methods section.

Page 4, paragraph 2.2. Cultivation of the CHO-K1-900 Cell Line and Purification of Chimeric mAb 900, lines 153-154.

Page 5, paragraph 2.2. Cultivation of the CHO-K1-900 Cell Line and Purification of Chimeric mAb 900, lines 156-168.

SEC Analysis and Aggregate Quantification. In direct response to your comment, we have performed analytical Size-Exclusion Chromatography (SEC) under non-denaturing conditions to assess the oligomeric state of the purified mAb 900. The SEC profile (now included as  Supplementary Figure S2) confirms that the main peak (>98% of the total area) elutes at a volume corresponding to the expected molecular weight of the non-reduced, dimeric scFv-Fc form (~100 kDa).

Page 5, paragraph 2.2. Cultivation of the CHO-K1-900 Cell Line and Purification of Chimeric mAb 900, lines 169-173.

Pages 16-17, paragraph 4.12. SEC, lines 606-613.

Supplementary material S2.

Thermal Stability Data. As suggested, we have included the thermal melting data obtained via circular dichroism spectroscopy. The determined melting temperature (Tm = 77 °C) indicates a stable, properly folded structure for the engineered mAb 900. This data is presented in a new panel and described in the text.

Page 5, paragraph 2.2. Cultivation of the CHO-K1-900 Cell Line and Purification of Chimeric mAb 900, lines 174-178.

Pages 16, paragraph 4.10. Determination of the Thermal Stability of mAb 900, lines 591-605.

Page 1. Abstract. Line 20.

Page 11, paragraph 3. Discussion, lines 321-330.

Supplementary material S3.

2. Regarding structural validation.

We thank the reviewer for highlighting the importance of structural validation. In this study, we prioritized a rigorous functional comparison to answer the primary question of whether chimerization impairs the antibody's activity. The fact that mAb 900 fully retains the high antigen-binding affinity and t virus-neutralizing activity of the parental murine 9E2 antibody provides the most direct and compelling evidence for the preservation of its functional epitope.

Importantly, we have successfully determined the three-dimensional structure of the antigen-binding fragment (scFv 900) in complex with the WNV DIII domain using X-ray crystallography and small-angle X-ray scattering method. These results allowed us to precisely localize the conformational epitope of this antibody.

However, this comprehensive structural study, including detailed epitope mapping and comparative analysis, constitutes a substantial body of work that is beyond the focused scope of the present manuscript. Therefore, these structural data will be presented and discussed in detail in a separate, forthcoming publication currently in preparation (tentatively titled "How does antibody 900 neutralize West Nile virus?").

In the revised Discussion section, we have added a statement acknowledging these completed structural findings.

Comments 4: The neutralization IC50 values for mAb 900 (74 µg/mL) and the murine 9E2 control (68.6 µg/mL) are considerably higher (weaker) than what is typically required for therapeutic antiviral antibodies. This limitation is not adequately acknowledged or discussed and raises concerns about the feasibility of clinical translation.

Response 4:

Thank you for pointing this out! In Table 3, there was an error in the units of measurement in the last column. It should be ng/mL instead of µg/mL, since dividing the initial antibody concentration (mg/mL) by the titer yields ng/mL. We have corrected the IC50 units in the table and in the text.

Page 7, paragraph 2.4. Virus-Neutralizing Activity of mAb 900, Table 3.

Page 7, paragraph 2.4. Virus-Neutralizing Activity of mAb 900, lines 224, 225, 228,229.

Page 11, paragraph 3. Discussion, lines 315, 316.

Comments 5: The BLI assays use few antibody concentrations and display noise and variability that undermine the reliability of KD calculations. ELISA results are presented as endpoint titers without binding curves, statistical analysis, or quantitative comparisons. Neutralization assays lack adequate controls, replicate numbers, and detailed reporting of infection conditions (e.g., MOI). Collectively, these issues reduce confidence in the data.

Response 5: We thank the reviewer for raising these methodological points. We have tried to take all these comments into account:

BLI Assays. The equilibrium dissociation constant (KD) was determined from measurements performed at multiple antibody concentrations, specifically chosen to include the low-nanomolar range critical for accurate fitting. The reported value is derived from a robust global fit of the complete dataset.

ELISA. The results are presented as binding curves (optical density vs. concentration). Student’s t-test was used for two-group comparisons.

Page 6, paragraph 2.3. Functional Activity of mAb 900 in ELISA, Figure 3, lines 189-193.

Page 7, paragraph 2.3. Functional Activity of mAb 900 in ELISA, Table 2, lines 195-198.

Neutralization Assays. The experiments included all necessary control groups (virus control, cell control). The assays were performed with an adequate number of replicates (detailed in the figure legends). The Materials and Methods section provides a comprehensive description of the infection conditions, including the cell line, multiplicity of infection (MOI), and incubation parameters.

Pages 7-8, paragraph 2.4. Virus-Neutralizing Activity of mAb 900, Table 3, lines 220-222.

Page 16. paragraph 4.10. Viruses, Cell Cultures and Neutralization Assay, lines 569-590.

We have further refined the text and figures to ensure these details are communicated with maximum clarity. We believe the presented data are robust and reliably support our conclusions regarding the preserved immunochemical properties of mAb 900.

Comments 6: The manuscript makes claims about preserved functionality and therapeutic promise that are not sufficiently supported by the limited dataset. Without in vivo validation, comprehensive Fc functionality testing, or pharmacokinetic studies, statements about therapeutic viability and extended half-life remain speculative.

Response 6: We thank the reviewer for this critical perspective. In the revised manuscript, we have significantly expanded the experimental dataset to provide stronger support for the functional assessment of mAb 900. Specifically, we have included new data from:

An in vivo protection study in a lethal mouse model of WNV infection, demonstrating significant protective efficacy (83.3% survival at the optimal dose).

An acute toxicity study in ICR mice, showing 100% survival and no observable adverse effects at the highest tested dose, supporting its preliminary safety profile.

These new results substantiate the high potential of mAb 900 as a candidate for further development.

We fully agree with the reviewer that comprehensive Fc-effector function profiling, detailed pharmacokinetic studies, and advanced preclinical trials are essential steps to unequivocally establish an antibody as a human therapeutic. Therefore, in the revised text, we have carefully rephrased our conclusions to avoid overstatement. Claims regarding "therapeutic viability" have been replaced with more precise language describing mAb 900 as a "promising preclinical candidate" or a "lead molecule for further investigation," with the newly acquired in vivo data forming the rationale for this assertion.

Page 1. Abstract. Lines 22-25.

Page 2. Introduction. Lines 83-85.

Page 18. 5. Conclusion. Lines 658-672.

Comments 7  Statistical methods are largely absent, aside from a brief mention of the Dunnett test. No discussion is provided on assay reproducibility or error reporting. Purity and yield claims are not quantitatively substantiated. Essential experimental controls are not described or shown.

Response 7: Thank you for your comment. We have added statistical data and control information to the ELISA, neutralization assay, section 4.15 Statistical Analysis in Materials and Methods.

Page 17, paragraph 4.15 Statistical Analysis in Materials and Methods, lines 652-656.

Round 2

Reviewer 2 Report

Comments and Suggestions for Authors

the authors have adequately addressed all concerns that have been raised. The manuscript has been significantly improved and can be accepted